# Functional Characterization of Rare Variants in OAT1/*SLC22A6* and OAT3/*SLC22A8* Urate Transporters Identified in a Gout and Hyperuricemia Cohort

**DOI:** 10.3390/cells11071063

**Published:** 2022-03-22

**Authors:** Jiří Vávra, Andrea Mančíková, Kateřina Pavelcová, Lenka Hasíková, Jana Bohatá, Blanka Stibůrková

**Affiliations:** 1Department of Cell Biology, Faculty of Science, Charles University, Vinicna 7, 128 43 Prague, Czech Republic; vavraji1@natur.cuni.cz; 2Department of Staphylococcal and Food-Borne Bacterial Infections, The National Institute of Public Health, Šrobárova 49/48, 100 00 Prague, Czech Republic; andrea.mancikova@szu.cz; 3Institute of Rheumatology, Na Slupi 450, 128 00 Prague, Czech Republic; pavelcova@revma.cz (K.P.); hasikova@revma.cz (L.H.); bohata@revma.cz (J.B.); 4Department of Pediatrics and Inherited Metabolic Disorders, First Faculty of Medicine, Charles University and General University Hospital, Ke Karlovu 455, 121 00 Prague, Czech Republic

**Keywords:** hyperuricemia, gout, OAT1, OAT3, urate transport

## Abstract

The OAT1 (*SLC22A6*) and OAT3 (*SLC22A8*) urate transporters are located on the basolateral membrane of the proximal renal tubules, where they ensure the uptake of uric acid from the urine back into the body. In a cohort of 150 Czech patients with primary hyperuricemia and gout, we examined the coding regions of both genes using PCR amplification and Sanger sequencing. Variants p.P104L (rs11568627) and p.A190T (rs146282438) were identified in the gene for solute carrier family 22 member 6 (*SLC22A6*) and variants p.R149C (rs45566039), p.V448I (rs11568486) and p.R513Q (rs145474422) in the gene solute carrier family 22 member 8 (*SLC22A8*). We performed a functional study of these rare non-synonymous variants using the HEK293T cell line. We found that only p.R149C significantly reduced uric acid transport in vitro. Our results could deepen the understanding of uric acid handling in the kidneys and the molecular mechanism of uric acid transport by the OAT family of organic ion transporters.

## 1. Introduction

Serum urate concentration is a complex phenotype influenced by both genetic and environmental factors, as well as the interactions between them. Hyperuricemia results from an imbalance between endogenous production and excretion of urate. Hyperuricemia is also a central feature in the pathogenesis of gout. The disorder progresses through several stages: asymptomatic hyperuricemia, acute gouty arthritis, intercritical gout, and chronic tophaceous gout. Not all individuals with hyperuricemia develop symptomatic gout, but the risk increases in proportion to the elevation of urate in circulation. The risk of gout development is conditioned not only by hyperuricemia but also by gender, weight, age, environmental and genetic factors, and their interactions.

Over the past decade, genome-wide association studies and meta-analyses have revealed over 30 common sequence variants influencing hyperuricemia or gout, mostly in urate transporters. The net amount of excreted uric acid is regulated mainly by urate transporters, such as organic anion transporter 1 (OAT1), organic anion transporter 3 (OAT3), urate transporter 1 (URAT1, a renal urate re-absorber) [1], solute carrier family two member 9 (SLC2A9, also known as glucose transporter member 9 (GLUT9)) [2], and ATP-binding cassette subfamily G member 2 (ABCG2, a high-capacity urate exporter expressed in the kidney and intestine) [3]. Dysfunction of URAT1 and GLUT9 reportedly causes inherited hypouricemia type 1 and type 2 [4], with high population specificity for dysfunctional variants in URAT1 [5,6,7]. The *ABCG2* gene is a well-established hyperuricemia/gout locus that plays a crucial role in renal and intestinal urate excretion; dysfunction of the ABCG2 is a risk factor for hyperuricemia and gout [8].

Most important for uric acid uptake is organic anion transporter 1 (OAT1 encoded by the *SLC22A6* gene) and organic anion transporter 3 (OAT3 encoded by the *SLC22A8* gene) [9,10,11]. OAT1 was discovered as an NKT (Novel kidney transporter) in the mouse kidney, in the proximal tubule [12]. In the same year, a homology transporter in rat kidneys was discovered [13]. The organic anion transporter in human kidneys is named OAT1. It was first discovered as a p-aminohippurate transporter [14]. OAT1 knocked out mice showed an increased accumulation of some organic compounds in the plasma, such as 3-hydroxybutyrate, 4-hydroxyphenyl lactate, 3-hydroxypropionate, benzoate, p-aminohippurate, and ochratoxin-A [15,16]. OAT1 was later found to be a uric acid transporter [17]. OAT1 transports uric acid into the cytoplasm of proximal epithelial cells and is essential for uric acid excretion, along with the other urate transporters [9,10]. Uric acids and other substrates are exchanged by antiport with α-ketoglutarate and other similar organic anions [13]. The Michaelis–Menten constant Km for uric acid transport is 943 ± 84 µmol/L [17]. OAT1 is localized in cell membranes probably as a homo-oligomer (homotrimer or with a higher number of subunits) [18]. In rodents, the transporter is localized on the basolateral membrane of the proximal tubule epithelial cells [12,13]. As in mice, OAT1 in human kidneys is localized in the basolateral membrane of epithelial cells of the proximal tubule but not in distal tubules, collecting ducts, or glomeruli [14]. Based on a computational prediction, the OAT1 protein has twelve transmembrane segments, with one large extracellular loop with an N-glycosylation site and two smaller intracellular loops [19]. Glycosylation of asparagine N39 was found in nascent proteins in cell membranes, but it was not shown to be necessary for the correct targeting of the protein. However, asparagine may be necessary for proper substrate binding [20]. The protein can be phosphorylated, and the phosphorylation of the serine residues in OAT1 expressed in LLC-PK1 cells leads to downregulation of p-aminohippurate transport activity [21]. The binding of substrate to the transporter is dependent on properties such as hydrophobicity, charge, or size [16].

Organic anion transporter three (OAT3) is coded by the *SLC22A8* gene, which is located on human chromosome 11. This transporter was first discovered as a product of the *Roct* gene in mice with an osteosclerosis mutation [11,22]. In humans, homologous transporter hOAT3 was discovered and expressed predominantly in the kidneys and weakly in skeletal muscle and the brain [11]. OAT3 was first identified relative to p-aminohippurate, taurocholate, estrone sulfate transport, predominantly seen in the kidneys and choroid plexus [23]. Other substrates include benzylpenicillin, 6-mercaptopurine, and antiviral drugs such as didanosine or zalcitabine [24,25]. The uptake of substrates is driven by exchange driven by the gradient of dicarboxylates, such as glutarate or α-ketoglutarate from the cytoplasmatic to the extracellular space [26,27]. In kidneys, OAT3, differentially from OAT1, is located on the basolateral membrane in proximal and distal tubules but not in the thin ascending limb and collecting duct [28]. In addition to the kidneys, it is also located in the retinal endothelium and choroid plexus; older works also report OAT3 mRNA in skeletal muscle [11,23,24]. The bioinformatics prediction model for OAT3 has eleven transmembrane spanning segments, one large and one small extracellular loop and one large and one small intracellular loop [19]. There are two putative glycosylation sites in the OAT3 structure, but the role of glycosylation, in this case, remains unclear. OAT3 is localized in the lipid rafts domain, where it interacts with associated proteins such as β-actin, caveolin 1, and myosin [29].

This study examined the role of five allelic variants in the OAT1 and OAT3 transporters, discovered in a hyperuricemia and gout cohort, on uric acid transport capacity in vivo and in vitro.

## 2. Materials and Methods

### 2.1. Subject

The analyzed set consists of a hyperuricemic group consisting of 36 subjects and a gout group consisting of 114 subjects. All patients were selected from the Institute of Rheumatology, Prague, the Czech Republic. A control group of 115 normouricemic subjects was selected from personnel working at the Institute of Rheumatology. In terms of serum uric acid (SUA), the definition of hyperuricemia was as follows: (1) men > 420 μmol/L (7.0 mg/dL) on two repeated measurements taken at least four weeks apart, and (2) women and children under 15 years > 360 μmol/L (6.0 mg/dL) on two repeated measurements taken at least four weeks apart.

We also examined a cohort of 34 pediatric patients with hyperuricemia and gout, of which ten individuals were also included, bringing the cohort up to 150 patients. We described this cohort in detail in our previous publication [30].

Gouty arthritis was diagnosed according to the American College of Rheumatology criteria: (1) the presence of sodium urate crystals seen in the synovial fluid using polarized microscopy or (2) at least six of 12 clinical criteria being met [31]. Patients suffering from secondary gout and other purine metabolic disorders associated with pathological concentrations of SUA were excluded. All tests were performed following standards set by the institutional ethics committees, which was approved on 30 June 2015 as project no. 6181/2015. All the procedures were performed following the Declaration of Helsinki.

### 2.2. Clinical Investigations

Urate and creatinine levels were measured using a Beckman Coulter AU system (Beckman Coulter, Brea, CA, USA). Metabolic investigations of purine metabolism (hypoxanthine and xanthine levels in urine) were also conducted using the method described in a previous study [32].

### 2.3. PCR Amplification and Sequencing Analysis

Genomic DNA of the Czech data set was extracted from whole blood drawn into EDTA tubes, as we described previously [33].

All coding regions and exon-intron boundaries in *SLC22A6* and *SLC22A8* were analyzed using genomic DNA, amplified using PCR, and purified as described previously [33]. The reference sequence for *SLC22A6* was defined as version ENSG00000197901, location Chromosome 11: 62,936,385–62,984,967, transcript version ENST00000377871.7. For *SLC22A8*, the reference genomic sequence was defined as version ENSG00000149452, location Chromosome 11: 62,989,154–63,015,841, transcript version ENST00000336232.7. The genotypes of allelic variants in the Czech control cohort were determined using PCR with allele-specific primers. Primer sequences and PCR conditions are available upon request. Predictions regarding the possible impact of non-synonymous allelic variants on protein function were determined using PolyPhen, Provean, Mutation Taster, SIFT, Human Splicing Finder, and MutPred predictive software.

### 2.4. Materials

The HEK293T cell line was purchased from Sigma (Cat. No.: 12022001-1VL). Cultivation medium DMEM, FBS, and supplements were purchased from Sigma Aldrich. Radiolabeled 14C uric acid MC-1394 was purchased from Hartmann analytic GmbH, Braunschweig, Germany. Plasmids with cloned *SLC22A6* WT (wild type) and *SLC22A8* WT pLenti-C-mGFP-P2A-Puro and mGFP primary antibody OTI2FG were purchased from Origene, Rockville, MD, USA. Mutagenesis was accomplished using Geneart Site-directed mutagenesis kits from Qiagen (Cat. No.: A13282), Hilden, Germany. Beta-actinin primary antibody was purchased from Cell Signaling (clone 8H10D10), Danvers, MA, USA. Rabbit anti-mouse HRP conjugated secondary antibody was provided by Invitrogen, cat. No.: A90-117P. Kits for plasmid DNA isolation were purchased from Qiagen, Hilden, Germany. Bradford assay kits were purchased from Biorad, Hercules, CA, USA. Cultivation plastic and were provided by VWR, Radnor, PA, USA. PVDF blotting membranes were purchased from Merck Immobilon (Cat. No. IPVH 07850). Other common chemicals came from Penta chemicals, Praha, Czech Republic or Sigma Aldrich, St. Louis, MO, USA.

### 2.5. Site-Directed Mutagenesis

We used plasmid pLenti-C-mGFP-P2A-Puro from Origene with *SLC22A6* (*OAT1*) or *SLC22A8* (*OAT3*) gene. To prepare allelic variants of these genes, we used Geneart site-directed mutagenesis kits. We designed the mutagenesis primers for each variant and performed mutagenesis PCR with template plasmid methylation. The primer used for site-directed mutagenesis is shown in Table 1. The reaction mixture included 4.0% DMSO and 0.3 mM MgSO_4_. After verifying the PCR product, a recombination reaction was carried out and further transformed the dH5alfa bacteria with a prepared plasmid for each variant. After plasmid amplification in bacteria, we isolate the plasmid DNA using Qiagen Maxi Kits. We verified the results by sequencing the gene’s coding sequence (Sequenation laboratory, Faculty of Science, Charles University, Prague, CZ), and this one was compared with the sequence published in the Ensembl database [34].

### 2.6. Cell Maintenance and Lipofection

Cell line HEK293T was cultivated in DMEM supplemented with 10% fetal bovine serum (FBS), gentamycin 0.04 mg/mL, 1× l-glutamine, and 1.0 mmol sodium pyruvate. Cells were frozen and grown to confluence in a T75 flask before the uptake study. Then, the cells were passaged with trypsin with EDTA and seeded into a 12-well dish previously treated with poly-d-lysine solution (0.1 mg/mL). The seeding density was 150 × 10^3^ cells per well were incubated until cca 80% confluence. At this time, the cells were transfected using lipofection with polyethyleneimine (PEI, total concentration 44.5 μg/mL) with a mixture containing serum-free DMEM medium and DNA. The DNA: PEI ratio was 1 ug plasmid DNA to 2.6 μg PEI. After 3.5 h of incubation, the lipofection mixture was flushed out and replaced with standard DMEM + 10% FBS. We prepared four wells of transfected cells for each variant (MOCK, WT, or allelic variant). The cells were incubated to confluency over a further 48 h and were then used for the uptake study.

### 2.7. Uptake Studies

For the uptake study, we used 30 μM radiolabeled uric acid (14C isotope) in a Hank balanced salt solution (NaCl 138.0 mM, KCl 5.0 mM; CaCl_2_ 1.0 mM, MgCl_2_ 0.5 mM, MgSO_4_ 0.4 mM; KH_2_PO_4_ 0.4 mM; NaHCO_3_ 4.0 mM; Na_2_HPO_4_ 0.3 mM; glucose 5.6 mM). The expression of the protein was verified in situ using a fluorescent microscope. First, the DMEM medium was removed, and the cells were preincubated with 500 μL HBSS without uric acid for 15 min at 37 °C. After this step, cells were incubated with uric acid in HBSS for 5 min at 37 °C; immediately after incubation, the dish was placed on ice, and the incubation solution was discarded. The cell monolayer was rinsed with ice-cold HBSS (3-times) with mild shaking on a shaker. The cells were lysed using 0.15 M NaOH on ice for 3 h, and the lysate was put into scintillation vials and neutralized with 1M HCl. We used a liquid scintillation cocktail, i.e., UltimaGold LSC cocktail (Sigma, Cat. No. L8286-SL). The activity of 14C uric acid was measured using a liquid scintillation PerkinElmer TriCarb 2900-TR scintillation counter (pre-measuring = 1 min, measuring = 5 min). Disintegrations per minute (DPM) were quenching corrected. An uptake assay was performed in three wells for each variant (MOCK, WT, or allelic variant). The fourth well was used for measuring the protein concentration. The same protocol was performed in this fourth well except for incubation with a uric acid isotope (HBSS only). Cells were lysed with 0.15M NaOH, the lysate was mixed with Bradford reagent, and the protein concentration was measured spectrophotometrically on a Nanodrop 2000.

### 2.8. Microscopy

The cells were seeded on cover glasses treated with poly-d-lysine (0.1 mg/mL) at a density of 100 × 10^3^ cells per glass. The cells were transfected using PEI lipofection (same protocol as above) with a 30% confluence. The cells were fixed using 4.0% paraformaldehyde in PBS (pH = 7.4) 48 h after transfection. Fixation time was 10 min (longer incubation times cause quenching of the mGFP signal). After fixation, the remaining paraformaldehyde was eliminated during incubation with NH_4_Cl. Cover glasses with cells were mounted in Mowiol with DAPI. The cells were observed using a Leica DM6 fluorescence microscope and LasX software for photography.

### 2.9. Immunoblotting

The cells were transfected in the same way described above (transfection at 80% confluence). The cell monolayer was harvested with a cell scraper, and after centrifugation in phosphate buffer saline (PBS), the cell pellet was dissolved in 200 μL 10% SDS (pH = 7.4) with 18 μL protease inhibitory cocktail (Sigma Cat. No.: I3786). The cell lysate was homogenized, on ice, in an ultrasound homogenizer UP50H (Hielscher Ultrasonics GmbH, Teltow, Germany) (25 impulses with 60% amplitude) and then mixed with 2x Laemmli buffer (Sigma Cat. No.: S34701-10VL). The samples were denatured at 56 °C or 25 °C for 60 min. The samples underwent SDS-PAGE with a 10% running gel and blotted on an Immobilon PVDF membrane. The membrane was blocked using 5% nonfat milk in PBS + 0.1% tween 20 for 90 min with mild shaking on a shaker. The membrane was incubated overnight at 4 °C with the primary antibodies against mGFP (Origene Cat. No.: OTI2FG, dilution 1:500) or beta-actinin (Cat. No.: 8H10D10, dilution 1:1000). After this step, the membrane was incubated with the secondary antibody conjugated with horseradish peroxidase (HRP, Sigma Cat. No.: A90-117P) 1:7000 dilute in 5% nonfat milk in PBS + 0.1% tween 20, at room temperature. Then, the membrane was incubated with a chemiluminescence substrate (Supersignal West Pico Plus (Thermo Scientific Cat. No.: 34577)), and the signal was detected using a Fuji LAS4000 CCD camera system.

### 2.10. Statistical Analysis

For evaluation of results of our uptake studies, we used unpaired, two-tailed Student´s *t*-test for acquiring *p*-value between wild type and allelic variant data set. *p* < 0.05 was considered as statistically significant. Error bars in graph represented standard deviation (S.D.). We perform statistical analysis in Microsoft Excel.

Clinical data were analysed by Fisher’s exact test (*p* < 0.05 was considered as statistically significant) and Wilcoxon rank-sum test (*p* < 0.005 was considered as statistically significant).

## 3. Results

### 3.1. Subjects

The subjects’ main demographic and biochemical characteristics are summarized in Table 2. Our cohort consisted of 114 individuals with primary gout (100 male/14 female), of which 35 patients (31% of the subcohort) had a positive family history for gout. In a sub-cohort of 36 hyperuricemia patients (21 male/15 female), seven patients (19% of the sub-cohort) had a positive family history.

### 3.2. Sequencing Analysis

In the *SLC22A6* gene, we identified one rare non-synonymous variant p.P104L (rs11568627) in our cohort of 150 patients. In the pediatric cohort, we identified the rare variant p.A190T (rs146282438). In the cohort of 150 patients, we also detected two common synonymous variants p.P84P (rs11568628) and p.P117P (rs11568629) and one rare synonymous variant p.Y111Y (rs756195374). Nine variants were found in the analyzed regions of introns, namely c.369+136C>T (rs10897312), c.797+8G>A (rs1323492446), c.798−57C>G (rs11568614), c.921+33C>T (rs2276300), c.922−28G>A (rs369866532), c.1252+61T>C (rs3017671), c.1361+56C>T (rs11568633), c.1361+25T>C (rs1031587383) and c.1362−172T>C (rs3017670)—Table 3.

In the *SLC22A8* gene, we detected three rare non-synonymous variants, p.R149C (rs45566039), p.V448I (rs11568486), and p.R513G (rs145474422). The p.V448I and p.R513Q variants were also found in the pediatric cohort. We further identified three synonymous variants, the rare p.L317L (rs57743826) and the common variants p.P51P (rs4149180) and p.T241T (rs2276299). In our cohort of 150 patients, we also detected two common intronic variants c.437+79G>C (rs4149182) and c.1326−62G>A (rs2187384). One patient was found to have intron variant c.728+133T>C, which was not listed in any of the genetic databases. Furthermore, another intronic variant, c.1326−4G>A (rs767079100), was detected; this variant may lead to alternative splicing—Table 3.

### 3.3. Uptake Studies

Transport capacity for both transporters and their allelic variants was examined. The HEK293T cell line was transiently transfected by lipofection with PEI. First, we examined the OAT1 transporter coded by the *SLC22A6* gene. We found two allelic variants in a cohort of rheumatologic patients; the first was an alanine by threonine substitution (p.A190T), and the second was a proline by leucine substitution (p.P104L). Both variants are located in the extracellular part of the protein—p.P104L in the large extracellular loop and p.A190T in the short linker between transmembrane helices (Figure 1D). In the uptake study, we measured the accumulation of 14C uric acid in HEK293T cells from the uptake of HBSS buffer during 5 min of incubation at 37 °C. We repeated the study two or three times. The uptake of uric acid we expressed as pmol of 14C uric acid per milligrams of total protein per one minute of incubation. For MOCK (untransfected) cells, the uptake was 1.89 pmol/mg·min. For cells transfected with the wild-type variant of the OAT1 protein, the uptake was 29.38 pmol/mg·min. For allelic variant p.A190T, the uptake was 25.47 pmol/mg·min, and, for p.P104L, the uptake was 30.98 pmol/mg·min. The standard deviation for each variant was calculated. Figure 1C plots the graph of uptake into each allelic variant. These results indicate that neither p.A190T nor p.P104L had any effect on uric acid uptake into HEK293T cells.

Second, we examined the OAT3 transporter, coded by the *SLC22A8* gene. For this gene, we found three SNP´s, each causing an amino acid substitution: arginine by cysteine (p.R149C), valine by isoleucine (p.V448I), and arginine by glutamine (p.R513Q). The p.R513Q variant is located in the extracellular C-terminus of the protein. Variants p.R149C and p.V448I are in the intracellular link between transmembrane helixes (Figure 2D). The same uptake studies and conditions for OAT1 were also used for the OAT3 transporter (5 min of incubation with 14C uric acid, at 37 °C). The experiment was repeated two or three times (for p.R149C). For MOCK, the measured uptake was 2.32 pmol/mg·min; for the wild type variant of OAT3, the uptake was 7.16 pmol/mg·min (4.1 times less than the OAT1 WT uptake). The uptake of p.R149C was significantly decreased and was 2.53 pmol/mg·min (35.3% of OAT3 WT uptake); for p.V448I, the uptake was 8.83 pmol/mg·min, and uptake of p.R513Q was 6.15 pmol/mg·min. The standard deviation for each variant was calculated. Figure 2C shows the uptake for each allelic variant. Our results indicate that allelic variant p.R149C had uric acid uptake activity at the level of untransfected HEK293T cells. Allelic variants p.V448I and p.R513Q had transport activity similar to the wild type variant of the transporter, but for these variants, the measurement of uric acid uptake had large deviations from the average value.

### 3.4. Microscopy

Localization of both the studied transporters and the variants was studied microscopically. The HEK293T cells were transiently transfected by lipofection in the same way as the uptake study. Each transporter and its variant were tagged using mGFP fluorescent protein on the C-terminus of the transporter. The cell nucleus was stained using 2-(4-aminophenyl)-1H-indole-6-carboxamidine (DAPI). The HEK293T cells expressing OAT1 WT showed transporter localization in the membrane. Both allelic variants, p.A190T and p.P104L, showed cytoplasmatic membrane localization (Figure 1A). For the OAT3 WT and its variants, membrane localization was examined in the same way. The localization of variant p.R149C, having transport capacity similar to untransfected cells (MOCK), was examined in detail. The cell membrane was stained with PKH26 stain, and OAT1 p.R149C showed membrane localization. Variants OAT1 p.V448I and p.R513Q were also localized on the cell membrane (Figure 2A).

### 3.5. Immunoblotting

We evaluated the expression of OAT1 or OAT3 fusion protein using mGFP. For OAT1 and its allelic variants p.A190T and p.P104L, we detected one major specific band of about 90 kDa and two smaller non-specific bands (Figure 1B). The molecular weight of the major band agrees with the calculated weight of the fusion protein (OAT1 61 kDa, mGFP 26 kDa, total 87 kDa). As the loading control, we used an antibody against CapZ (β-actinin) protein, giving one band with a molecular weight of 42 kDa. The density of all bands was the same for OAT1 WT, p.A190T, and p.P104L. This fact suggests the same level of protein expression for all variants. Immunoblotting with OAT3 WT, p.R149C, p.V448I, and p.R513Q fusion protein (Figure 2B) was done similarly. We detected a major band with a molecular weight of 90 kDa and one smaller band. The density of all bands was similar; CapZ (β-actinin) was the loading control. The expression of OAT1 transporter protein and its variant was at the same level.

### 3.6. Evolutionary Comparison

We compared amino acid sequences of OAT1 and OAT3 across mammalian species using the UniProt Align tool [35]. In this comparison, we focused on amino acids, which were substituted by the researched allelic variants. For OAT1, we found that proline 104 was conserved among all selected mammalian species. However, proline substitution by leucine (p.P104L) produced no significant change in transport capacity in our in vitro study. Substitution of alanine by threonine (p.A190T) was not conserved among rodent mammals, such as mice (*Mus musculus*) or rats (*Rattus norwegicus*). In these species, alanine is substituted by noncharged and nonpolar valine or isoleucine (Figure 3A).

For OAT3, we found that arginine 149 is evolutionarily conserved among all selected mammalian species and may be crucial for transport activity. Our in vitro uptake study showed that the p.R149C substitution significantly decreased uric acid uptake into HEK293T cells. Valine 448 is conserved only in chimpanzees (*Pan troglodytes*) and macaques (*Maccaca mulata*), while in other species, it is substituted by alanine (*Sus scrofa*) or isoleucine (*Mus musculus*, *Rattus norwegicus*, and *Bos taurus*). Arginine 513 is conserved among all selected species except *Mus musculus* and *Rattus norwegicus*; in these two, arginine is substituted by glutamine (Figure 3B).

## 4. Discussion

Our recent paper investigates the role of allelic variants causing amino acid substitutions in the OAT1 and OAT3 transporter. These variants were obtained by sequencing a cohort of 150 patients from the Institute of Rheumatology.

For OAT1, we found two substitutions of amino acid residues in the patient cohort: alanine by threonine (p.A190T) and proline by leucine (p.P104L). We did not find a significant difference in transport capacity between these two allelic variants. Both variants were mainly expressed on the plasma membrane of HEK293T cells. Similarly, we found the same level of expression of the OAT1-mGFP fusion protein. Allelic variant p.P104L was located in the large extracellular loop of the protein, while p.A190T was located at the end of the link between the 3rd and 4th transmembrane segments. The p.P104L substitution was a nonpolar amino acid by another nonpolar amino acid, but the p.A190T substitution involved a nonpolar alanine substituted by a polar threonine. Interestingly, the p.A190T substitution minimally affected uric acid transport. Neither (p.A190T and p.P104L) appeared crucial for uric acid transport. Another substitution in the large extracellular loop, which involved arginine 50 being replaced by histidine (p.R50H), decreased uptake of antiviral drugs such as adefovir. This arginine is found in an evolutionarily conserved motif [36]. This first extracellular loop is conserved among organic anion and cation families [37]. The influence of this allelic variant on the transport of methotrexate, ochratoxin, and p-aminohippurate was investigated using the Xenopus oocyte expression system. For ochratoxin, uptake was increased but not significantly [38]. Proline 104 is conserved among higher vertebrates over alanine 190, not conserved in two rodent species (mouse and rat). In this context, the alanine residue seems less important. It is possible that the link between the 3rd and 4th transmembrane segments has only a connecting function and has no function in uric acid transport or protein targeting. In contrast, the link between transmembrane segments 2 and 3 is evolutionarily conserved and forms the MFS motif [37].

In the OAT3 transporter, we found three substitutions of amino acids in the patient cohort: arginine by cysteine (p.R149C), valine by isoleucine (p.V448I), and arginine by glutamine (p.R513Q). The transport capacity of the p.R149C allelic variant was significantly decreased and was similar to the control. Like the other two, this variant was located on the plasma membrane. The other two variants had similar transport capacities as the wild type, but the standard deviations were substantial. This could have been due to the different quality of the cell aliquot used and the loss of cell lysate during handling. All variants and the wild type had the same expression level. Of these three amino acid residues, only arginine 149 was conserved in higher vertebrates. Arginine 513 was not conserved in rodents (mouse and rat), but valine 448 was conserved in apes. It is, therefore, possible that these two residues are not crucial for uric acid handling by this transporter. A potential mechanism for rapidly aborting uric acid transport is substitution of basic arginine by a polar, but uncharged, cysteine. Arginine 149 is located in the intracellular loop between helices 2 and 3. Similar substitution (polar serine instead of basic arginine) significantly decreases the uptake of cimetidine and estrone sulphate [39]. Thus, arginine 149 is crucial for the transport of uric acid and other compounds. It seems that more polar molecules (estrone sulfate or uric acid) bind tightly to the arginine residue compared to less polar amino acids, such as cimetidine. Of course, further investigation is needed to explain in detail the influence of arginine 149. Furthermore, arginin 149 and proline 104 are conserved in primary structure of both transporters OAT1 and OAT3 (data not shown), but it is surprising that only R149 has a crucial role for uric acid uptake. It is possible that proline 104 and the surrounding region have no role in the binding of molecules of uric acid and/or cotransported dicarboxylates. However, this hypothesis is necessary to that hereafter tested. Interestingly, valine 448 is substituted by isoleucine, which is usually present at this position in rodents and cows. Similarly, arginine 513 is usually substituted by glutamine in rodents. Together with the structural similarity between valine, alanine, and isoleucine, these facts may explain why uric acid uptake is similar to that of the wild type (except for the large standard deviation). Substitution of valine 448 by isoleucine caused a non-significant decrease in cimetidine and estrone sulfate transport [39]. Similarly, the substitution of arginine 513 by glutamine naturally occurs in rodents. Interestingly, the replacement of basic arginine with polar glutamine only caused a small, non-significant decrease in uric acid transport. A possible explanation is that the role of p.R513Q and the small extracellular loop is not crucial in uric acid transport.

Both of these transporters are located on the basolateral side of epithelial cells, and both influence uric acid uptake into the cell from the blood pole [10,14,17,28]. All the detected allelic variants caused hyperuricemia or gout (Table 4). It is clear why the substitution of arginine 149 by cysteine caused gout since the amount of uric acid transported by this OAT3 variant was reduced (Figure 2C). The question is why other OAT3 transporters have no compensatory effect on urate uptake into the cell. The gout symptoms in this patient may have been caused by dysfunction of the *ABCC4* apical efflux transporter (Table 5). The other two allelic variants of OAT3, p.V448I and p.R513Q, showed no significant decrease in urate uptake in vitro. However, the patients with these polymorphisms had a homozygous mutation in transporters *ABCC4*, *SLC17A3*, or *SLC17A1*. All these transporters are located on the apical side of the proximal tubule epithelial cells, where they excrete uric acid into urine [9]. Allelic variants of OAT1 p.A190T and p.P104L had a clinical manifestation but had no significant effect on in vitro uptake. A possible explanation could be a homozygous or heterozygous mutation in transporters *SLC17A1*, *SLC17A3*, *ABCG2*, or *ABCC4*. In our previous study, we show that *ABCG2* variants (especially p.Q141K variant) are crucial for the onset of hyperuricemia and gout, namely in pediatric-onset patients [40,41,42]. On the other hand, we found out that allelic variants in *SLC2A9* (GLUT9) identified in our cohort had no effect on uptake of uric acid [43]. Now, we examine presence and impact of selected allelic variants in other urate transporters in hyperuricemia/gout subjects (Table 5). These apical membrane-localized transporters are responsible for the excretion of uric acid in the urine [9]. Allelic variant p.I269T in *SLC17A1* increases the transport capacity for uric acid [44]. This fact may explain why homozygous patients (patient No. 3 and 7, Table 5) only had hyperuricemia and not fully developed gout. The p.G279R substitution in *SLC17A3* caused a decrease in uric acid excretion, while the p.A100T variant had no significant effect [45]. We have no clinical or experimental data, which should elucidate the relation between *SLC22A12* (URAT 1), the main transporter which uptakes urate from urine and *SLC22A6* and/or *SLC22A8*, and its allelic variants. It is possible that the effect of variants, which decreases transport in OAT1 and OAT3 (p.R149C), may be compensated by the effect loss of function variants in *SLC22A12* because these transporters work antagonistically.

Overall, these data indicate the ambiguous clinical effect of the identified variants on FE-UA. The age range of subjects with the identified variants (from 11 to 74 years) may have influenced the levels of SUA and FE-UA: the SUA levels are low in infancy (131–149 µmol/L; 2.2–2.5 mg/dL at 2–3 months of age) due to the high FE-UA levels (>10%); the FE-UA levels decrease to approximately 8% at age one and then stay through childhood, which is associated with mean SUA levels (208–268 µmol/L; 3.5–4.5 mg/dL) of children. At adolescence (after age 12), the FE-UA levels significantly decrease in boys but not in girls, resulting in a further significant increase in SUA levels in young men but not in young women [46].

The main limitation of this study is using the cell line model that overexpresses only one transporter. This model of uptake assay does not include physiological interaction between different transport proteins for uric acid. This shortage may be diminished by validating it with some in vivo assay in the future, for example by using a genetic modified mouse with oxonate treatment.

## 5. Conclusions

We found five rare allelic variants in a cohort of 114 gout and 36 hyperuricemia patients. Two non-synonymous variants p.A190T and p.P104L were found in OAT1. Three non-synonymous variants p.R149C, p.V448I, and p.R513Q were identified in OAT3. Of the identified variants, only the arginine 149 substitution with cysteine significantly decreased uric acid uptake in the in vitro study. Our findings might support a “Common Disease, Multiple Common and Rare variant” hypothesis for the association between urate transporters and hyperuricemia/gout susceptibility in a European population. Additionally, these findings of population-specific genetic variants could deepen our understanding of the heritability of SUA levels and hyperuricemia/gout.

## Figures and Tables

**Figure 1 cells-11-01063-f001:**
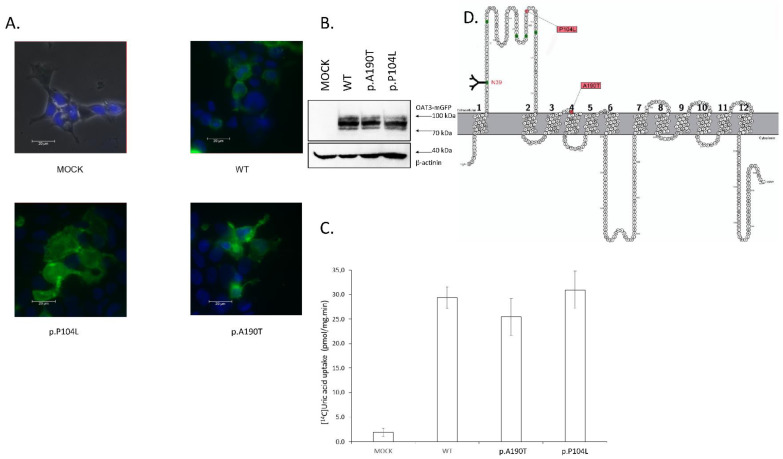
(**A**) Expression of *SLC22A6* (OAT1) and it´s allelic variants in HEK293T cells line. Wild type protein was tagged with C-terminal GFP tag and cells was transiently transfected by polyethyleneimine lipofection. Samples was fixed by 4% paraformaldehyde and picture was take by Leica DM6 microscope in 400× magnification. Plasmatic membrane localization was observed in all variants. Cell nucleus was stain by DAPI (blue). (**B**) Immunoblotting of HEK293T transfected by OAT3 WT and its allelic variant. Fusion protein OAT3-mGFP had majority band with molecular weight 90 kDa. CapZ (β-actinin) protein is used as loading control and it had one major band with 42 kDa molecular weight. (**C**) Transport of ^14^C radiolabeled uric acid by HEK293 cells transfected by OAT1 WT and its variant. The data are shown as pmol uric acid transported into the cytoplasm per one minute and total protein. Data are expressed with ±SD, *n* = 2. Statistical analyses for significant differnces was calculate by Students *t*-test (**D**) Topological model of human OAT1 protein (encoded by gene *SLC22A6*). Allelic variants p.P104L and p.A190T (single amino acid exchange) are colored by pink color. Asparagine 39 is N-linked glycosilated, other putative glycosylation motifs are colored by green. Edited on the base [19].

**Figure 2 cells-11-01063-f002:**
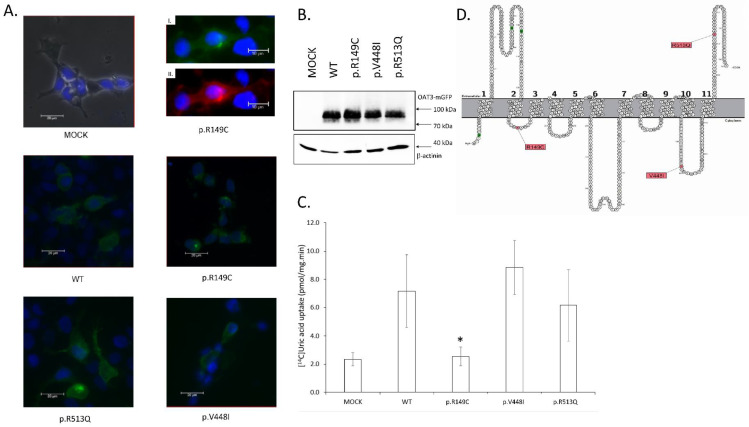
(**A**) Expression of *SLC22A8* (OAT3) and it´s allelic variants in HEK293T cells line. Wild type protein was tagged with C-terminal GFP tag and cells was transiently transfected by polyethyleneimine lipofection. Samples was fixed by 4% paraformaldehyde and picture was take by Leica DM6 microscope in 400× magnification. Plasmatic membrane localization was observed in all variants. Cell nucleus was stain by DAPI (blue). Higher magnification of allelic variant R149C of transporter OAT3 C-terminus tagged with GFP. Protein was expressed in HEK293 cell line. Cell nucleus was stain with DAPI. **I.** Fluorescence signal of GFP tagged protein. **II.** Cytoplasmatic membrane staining by PKH26 stain. (**B**) Immunoblotting of HEK293T transfected by OAT3 WT and its allelic variant. Fusion protein OAT3-mGFP had majority band with molecular weight 90 kDa. CapZ (β-actinin) protein is used as loading control and it had one major band with 42 kDa molecular weight. (**C**) Transport of ^14^C radiolabeled uric acid by HEK293 cells transfected by OAT3 WT and its allelic variants. The data are shown as pmol uric acid transported into the cytoplasm per one minute and per total protein in cell lysate. Data are expressed with ±SD, *n* = 2. Statistical analyses for significant differnces was calculate by Students *t*-test (*, *p* < 0.05). (**D**) Topological model of human OAT3 protein (encoded by gene *SLC22A8*). Allelic variants p.R149C, p.V448I and p.R513Q (single amino acid exchange) are colored by pink color. Putative glycosylation motifs are colored by green (for OAT3 is no experimental evidence for glycosylation). Edited on the base [19].

**Figure 3 cells-11-01063-f003:**
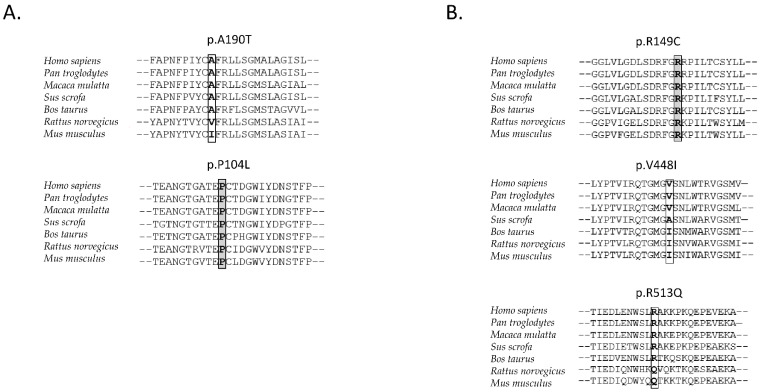
(**A**) OAT1 (*SLC22A6*) evolutionarily conserved amino acids among seven mammalian species. The position of non-synonymous substitution among seven species is indicated with grey fill. Identity with human OAT1 *Pan troglodytes* 97.3%, *Macaca mulatta* 94.7%, *Sus scrofa* 86.5%, *Bos taurus* 84.2%, *Rattus norvegicus* 85.9%, and *Mus musculus* 83.3%. (**B**) OAT3 (*SLC22A8*) evolutionarily conserved amino acids among seven mammalian species. The position of non-synonymous substitution among seven species is indicated with grey fill. Identity with human OAT3: *Pan troglodytes* 99.4%, *Macaca mulatta* 96.3%, *Sus scrofa* 81.7%, *Bos taurus* 81.7%, *Rattus norvegicus* 78.8%, and *Mus musculus* 77.5%, performed by using the UniProt Align tool [35].

**Table 1 cells-11-01063-t001:** Primers for site-directed mutagenesis of allelic variants in *OAT1* and *OAT3*.

Allelic Variant	Forward (5′ to 3′)	Reverse (5′ to 3′)
OAT1 p.P104L	GGGCCACAGAGCTCTGCACCGATGGCTGGATC	GATCCAGCCATCGGTGCAGAGCTCTGTGGCCC
OAT1 p.A190T	CCCATCTACTGCACCTTCCGGCTCCTCTCGG	CCGAGAGGAGCCGGAAGGTGCAGTAGATGGG
OAT3 p.V448I	CAAACAGGTATGGGCATAAGTAACCTGTGGAC	GTCCACAGGTTACTTATGCCCATACCTGTTTG
OAT3 p.R513Q	GAAAACTGGTCCCTGCAGGCAAAGAAGCCAAAG	CTTTGGCTTCTTTGCCTGCAGGGACCAGTTTTC
OAT3 p.R149C	GTCTGACAGGTTTGGCTGCAGGCCCATCCTGAC	GTCAGGATGGGCCTGCAGCCAAACCTGTCAGAC

**Table 2 cells-11-01063-t002:** The main demographic and biochemical characteristics of the subjects.

Characteristic		All Patients			Gout Patients			Hyperuricemia Patients			*p*-Value
		(*N* = 150)			(*N* = 114)			(*N* = 36)			#
		*N*	%		*N*	%		*N*	%		
Gender	Male	121	80.7		100	87.7		21	58.3		0.0004
	Female	29	19.3		14	12.3		15	41.7		
Familial occurrence		42	28		35	30.7		7	19.4		0.2097
			(29 *)						(22.6 *)		
**Characteristic**		** *N* **	**Median**	**Range**	** *N* **	**Median**	**Range**	** *N* **	**Median**	**Range**	** *p* ** **-Value**
			**(IQR)**			**(IQR)**			**(IQR)**		**†**
Age at examination [years]		150	59	3–80	114	59	30–80	36	55.5	3–78	0.13335
			21			19			34.3		
BMI at examination		108	29.1	16–43.4	80	29	20.6–43.4	28	30.1	16–37.5	0.7391
			5.3			5.4			5.1		
Gout/hyperuricemia onset [years]		125	45	13–77	112	45	18–77	13	48	13–70	0.4025
			23			21.3			36		
SUA at examination, with medication [µmol/L, (mg/dL)]		137	373(6.2)	163–725 (2.7–12.1)	107	371(6.2)	163–725 (2.7–12.1)	30	411(6.9)	240–628 (4.0–10.5)	0.1197
			123(2.1)			118(1.9)			155.3(2.5)		
FEUA at examination, with medication		135	3.4	0.9–11.8	107	3.4	0.9–11.8	28	3.5	1.3–8.4	0.974
			2.1			2			2.2		

# Fisher’s exact test (significance level: 0.05). † Wilcoxon rank-sum test (significance level: 0.005). * relative frequencies when missing information about familial occurrence was excluded. IQR = interquartile range; BMI = body mass index; SUA = serum urid acid; FEUA = fractional excretion of urid acid.

**Table 3 cells-11-01063-t003:** Non-synonymous allelic variants in *SLC22A6* and *SLC22A8* were identified in a cohort of 150 patients with primary hyperuricemia and gout.

Gene	Reference SNP Number	Position CDS	Position aa	Variant Allele Hetero/Homozygotes	Allelic Variant MAF	Normo-Uricemia Control MAF	European MAF
*SLC22A6*	rs11568627	c.311C>T	p.P104L	2/0	0.007	0.01	0.005
*SLC22A6*	rs146282438	c.568G>A	p.A190T	1/0	0.004	0	0
*SLC22A8*	rs45566039	c.445C>T	p.R149C	1/0	0.003	0	0
*SLC22A8*	rs11568486	c.1342G>A	p.V448I	2/0	0.007	0.005	0.015
*SLC22A8*	rs145474422	c.1538G>A	p.R513Q	2/0	0.007	0.01	0.003

**Table 4 cells-11-01063-t004:** Main demographic and biochemical characteristics of patients with non-synonymous variants in the *SLC22A6* and *SLC22A8* genes. GFR was estimated via the CKD-EPI method.

Variants in *SLC22A6*/*SLC22A8* (Patient Identification)	Gender	Diagnosis	Familial Occurrence	Age at Examination [years]	Gout/ Hyper- Uricemia Onset [years]	BMI at Examination	SUA without Medi- cation [µmol/L]	SUA with Medi- cation [µmol/L]	FEUA without Medi- cation	FEUA with Medi- cation	GFR (mL/ min/ 1.73 m^2^)
p.P104L (patient 1)	male	gout	no	60	60	N/A	N/A	451	N/A	7.4	N/A
p.P104L (patient 2)	male	hyperuricemia	yes	59	18	30.4	492	419	5.2	5.5	72
p.A190T (patient 3)	male	hyperuricemia	yes	11	10	20	N/A	367	N/A	4.9	N/A
p.R149C (patient 4)	female	gout	no	73	73	32.3	683	N/A	0.8	N/A	47
p.V448I (patient 5)	male	gout	no	30	30	30.8	587	406	3	2.1	102
p.V448I (patient 6)	male	hyperuricemia	N/A	48	N/A	N/A	N/A	426	N/A	3.3	N/A
p.V448I (patient 7)	male	hyperuricemia	yes	14	13	24.3	N/A	435	N/A	3.7	N/A
p.R513Q (patient 8)	male	gout	no	74	41	29.7	484	373	4.2	3.4	79
p.R513Q (patient 9)	female	hyperuricemia	yes	11	6	25.5	433	N/A	4.6	2.1	146
p.R513Q (patient 10)	male	hyperuricemia	yes	14	14	21.8	371	N/A	3.6	N/A	150
p.R513Q (patient 11)	male	gout	no	34	25	N/A	N/A	284	N/A	2.3	N/A

**Table 5 cells-11-01063-t005:** Overview of genetic variants in other genes encoding urate transporters in patients with non-synonymous allelic variants in the *SLC22A6* and *SLC22A8* genes. wt = wild type, HT = heterozygous variant, HM = homozygous variant.

Variants in Other Genes	Gene	*ABCG2*	*SLC2A9*	*SLC2A9*	*SLC2A9*	*SLC2A9*	*SLC2A9*	*SLC22A13*
	Reference SNP Number	rs2231142	rs2276961	rs16890979	rs73225891	rs2280205	rs6820230	rs72542450
	AA Change	p.Q141K	p.G25R	p.V282I	p.D281H	p.P350L	p.A17T	p.R16H
**variants in** ***SLC22A6*/** ** *SLC22A8* **	p.P104L (patient 1)	HT	wt	HT	HT	wt	wt	wt
	p.P104L (patient 2)	HT	HT	wt	wt	HT	wt	wt
	p.A190T (patient 3)	wt	HM	wt	wt	HT	wt	wt
	p.R149C (patient 4)	wt	HM	wt	wt	HM	wt	HT
	p.V448I (patient 5)	wt	HT	HT	wt	HT	wt	wt
	p.V448I (patient 6)	wt	HT	HT	wt	wt	wt	wt
	p.V448I (patient 7)	wt	HM	wt	wt	HM	wt	wt
	p.R513Q (patient 8)	wt	wt	wt	wt	wt	HM	wt
	p.R513Q (patient 9)	HT	HM	wt	wt	HM	wt	wt
	p.R513Q (patient 10)	wt	HM	wt	wt	HM	wt	wt
	p.R513Q (patient 11)	HT	HT	wt	wt	HT	wt	wt
**variants in other genes**	**Gene**	** *SLC17A1* **	** *SLC17A3* **	** *SLC17A3* **	** *SLC17A3* **	** *ABCC4* **	** *ABCC4* **	** *ABCC4* **
	**Reference SNP Number**	**rs1165196**	**rs56027330**	**rs1165165**	**rs56027330**	**rs2274406**	**rs1678339**	**rs1751034**
	**AA Change**	**p.T269I**	**p.G70R**	**p.A100T**	**p.G279R**	**p.R317S**	**p.L904F**	**p.K1116N**
**variants in** ***SLC22A6*/** ** *SLC22A8* **	p.P104L (patient 1)	HT	wt	HT	HT	HT	wt	HM
	p.P104L (patient 2)	wt	wt	HM	HM	HT	HM	wt
	p.A190T (patient 3)	HM	HT	wt	wt	HM	HM	wt
	p.R149C (patient 4)	HT	wt	wt	wt	HT	HM	wt
	p.V448I (patient 5)	wt	wt	HM	HT	wt	HM	wt
	p.V448I (patient 6)	HT	wt	wt	wt	HM	wt	HM
	p.V448I (patient 7)	HM	HT	wt	wt	HM	HM	wt
	p.R513Q (patient 8)	HT	wt	HT	wt	HT	HM	wt
	p.R513Q (patient 9)	wt	wt	wt	wt	HM	HM	wt
	p.R513Q (patient 10)	HT	HT	HT	wt	wt	HM	wt
	p.R513Q (patient 11)	wt	wt	wt	wt	HT	wt	HT

## Data Availability

The data presented in this study are available on request from the corresponding author.

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
