# Peer review of "Functional Characterization of Rare Variants in OAT1/SLC22A6 and OAT3/SLC22A8 Urate Transporters Identified in a Gout and Hyperuricemia Cohort"

_cells, 2022, doi:10.3390/cells11071063_

Round 1
Reviewer 1 Report
Check the reference list and conform to the ICMJE suggestions if not otherwise requested by the Editorial Board.
Author Response
Dear Reviewer,
Thank you very much for your time spended with our manuscript and your report.
Comment:
Check the reference list and conform to the ICMJE suggestions if not otherwise requested by the Editorial Board.
Reaction:
Thank you for you comment. We align our references in accordance with instruction of Editorial.
With best regard
Jiří Vávra
Reviewer 2 Report
Overall summary
This study identified some rare variants in both OAT 1 OAT3 in a cohort of a European population cohort. Specifically, the non-synonymous variants p.R149C in OAT3 was significantly associated with uric acid uptake in HEK293T cell line. Interesting work but replication is needed given the specificity of SLC22A8 in uric acid is very low compared with SLC22A12.
Specific Comments
- Page 2 line 49. There is a typo. It should read two A 9, not twenty-two A 9.
- Please define abbreviations noted in tables. For example, MAF, aa, SNP, and CDS
- Table 5- it’s unclear why the authors targeted these specific SNPs among all the other non-synonymous SNPs.
- There was no statical analyses section within the article. The authors didn’t provide a robust statical assessment of the uptake studies. Please provide a detailed approach for the statical analyses to compare uric acid uptake between different cell lines.
- Please provide uric acid measurement in mg/dL in table 2.
- Please provide information on the estimated glomerular filtration rate (eGFR), smoking history, and alcohol use if available.
- Please provide uric acid measurement in mg/dL in the materials and methods section when defining hyperuricemia.
- Given the biological and clinical importance of SLC22A12 in uric handling, it’s important to assess how the novel variants in SLC22A6 and SLC22A8 interact with known loss of function variants in SLC22A12
- Assessing linkage disequilibrium between these rare variants is warranted. There is a high degree of homology between SLC22A6 and SLC22A6, which may explain why only one nonsynonymous SNP (p.R149C in OAT3) showed an association with uric acid uptake.
- A discussion of the study limitations is also needed.
General Comments
- The discussion section is so dense and very difficult to read. Please break the discussion sections into smaller paragraphs.
